# The Complex Nature of the Abnormally Weak Absorption of Cosmic Ray Hadrons in Lead Calorimeters at Super-High Energies

Alexander S. Borisov[1*], Evgeniya A. Kanevskaya[1], Mikhail G. Kogan [1], Rauf A. Mukhamedshin [2], Vitalyi S. Puchkov[1] and Shakarmamad G. Yormamadov [3]

**1** P.N. Lebedev Physical Institute of Russian Academy of Sciences, Moscow, 119991 Russia
**2** Institute for Nuclear Research of Russian Academy of Sciences, Moscow, 117312 Russia
**3** S. Umarov Institute of Physics and Technology, National Academy of Sciences of Tajikistan, Tajikistan

⋆ asborisov55@mail.ru

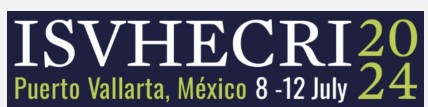

*22nd International Symposium on Very High Energy Cosmic Ray Interactions (ISVHECRI 2024)
Puerto Vallarta, Mexico, 8-12 July 2024*

## Abstract

This paper presents the joint analysis of the results of two similar experiments on the absorption of cosmic ray hadrons using deep lead calorimeters (namely, X-ray emulsion chambers) with large air gap, which were exposed at high altitudes in the Tien Shan and Pamirs mountains. It was found that Monte Carlo simulation of the both experiments allows to reproduce most of specific features of the experimental absorption curve, namely the position, amplitude and width of the peak of electromagnetic origin observed beyond the air gap, assuming that the value of cross section for the production of charm hadrons is as high as $\sigma_{pp \to c\bar{c}} \sim 8$ mb at $\langle E_{Lab} \rangle \sim 75$ TeV and at $x_{Lab} \gtrsim 0.01$. However, we observe at extreme depths of both calorimeters a significant excess of blackening spots (charged particle tracks) compared to the simulations, which cannot be explained by charmed particle production alone. New factors could be carefully considered, such as dark photons, which are currently being searched for by the NA62 collaboration at CERN, or strangelets.

# 1   Introduction

An abnormally weak absorption of cosmic ray hadrons in lead targets was first observed in 1973-1974 in an experiment with Big Ionization Calorimeter (BIC) at the Tien Shan Mountain Research Station (TSS) [1, 2]. The total thickness of lead plates in BIC of 36 m$^2$ in area was 75 cm ($\sim$ 850 g/cm$^2$). Just after the exposure of BIC, a hypothesis was put forward [3] about the existence of some unknown long-flying hadronic component (LFC) of cosmic rays. Soon after the BIC exposure and the discovery of charmed hadrons at accelerators (first with hidden charm in 1974 and then with open one in 1976) it was suggested that the LFC might be represented by charmed particles [4]. Since then some competing hypotheses have been put forward to explain the LFC nature, for instance, manifestation of quark strange matter in the form of production of strangelets by strange quark stars [5] or emission of bundles of high-energy direct muons generated by charmed hadrons in the atmosphere [6].

A similar phenomenon was observed in the Pamirs in the 1980s in X-ray emulsion chamber (XREC) experiment, where deep uniform lead chambers 110 cm thick were exposed for several years [7]. It was found out that it is impossible to fit the experimental absorption curve by one exponential law. At small depths (t < 78 radiation lengths) experimental points obeyed exponential with absorption length $\Lambda \sim 210$ g/cm$^2$, while at greater depths ($t > 78$ r.l.) it was almost 310 g/cm$^2$. To proof the hypothesis that excessive cascades are initiated by charm particles, a new two-tier emulsion chamber calorimeter with a large air gap, namely 2.5 m, was designed [8]. The size of the air gap was chosen so that charmed particles could effectively decay within it. Taking into account the life-time of the D mesons ($\sim 10^{-13}$ s) and their relativistic Lorentz-factor $\gamma$, we get the optimal value for $H = c\tau\gamma = c\tau E/m \approx 2.5$ m, where $E$ and $m$ are the energy and mass of charmed hadrons.

Since charmed particles decay predominantly through electromagnetic channels, they can be expected to appear as additional electromagnetic cascades in the upper part of the lower block of the two-tier lead calorimeter. Accordingly, such additional electromagnetic cascades should form a peak on the absorption curve of hadrons.

# 2   Experiment and Simulation

Finally, after several attempts to realize the designed experiment, it was performed first at the Tien Shan and then at the Pamirs. In this paper we present the results of a joint analysis of data from two close experiments in which the same type of two-tier lead X-ray emulsion calorimeters with a large air gap were exposed at mountain heights. In the experiment carried out in 2007 – 2008 at the TSS at an altitude of 3,340 m above sea level, an XREC with an area of $S = 48$ m$^2$ and an air gap of $H = 2.16$ m with a total thickness of the lead absorber of 50 cm was used, and in the experiment conducted in 2011 – 2012 at the Pamirs at an altitude of 4,370 m a.s.l., an XREC with an area of $S = 36$ m$^2$ and a gap of $H = 2.5$ m was used, with a lead-absorber total thickness of 67 cm.

The area of the upper block of both chambers was slightly larger than that of the lower

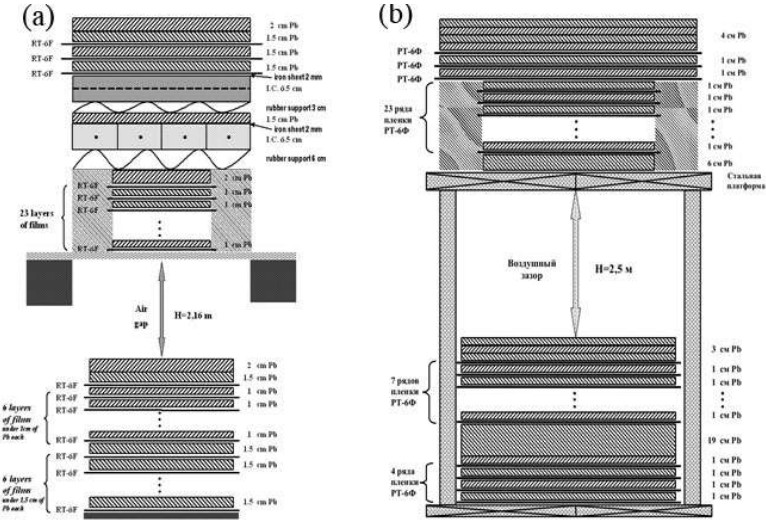

Figure 1: Schemes of two-tier XRECs at (a) the TSS (3,340 m a.s.l., $H = 2.16$ m, $S_{up} = 48$ m$^2$, $S_{low} = 32$ m$^2$, $E_\gamma^{thr} \approx 6$ TeV) and (b) the Pamirs (4,370 m a.s.l., $H = 2.5$ m, $S_{up} = 36$ m$^2$, $S_{low} = 24$ m$^2$, $E_\gamma^{thr} \approx 6$ TeV).

49  one (32 m$^2$ and 24 m$^2$, respectively), which made it possible to cut off background particles
50  with large zenith angles arriving at the lower block of the installation bypassing the upper one.
51  Background suppression was also achieved due to the fact that the assembling of the chambers
52  began from the upper block, while disassembling always started from the lower block.
53      The detailed description of the experiments with two-tier XRECs as well as their some pre-
54  liminary results can be find in [9]. To analyze the data from both experiments, detailed Monte
55  Carlo simulation of the development of hadron cascades in XRECs, as well as the response of
56  the XRECs were carried out employing FANSY 1.0 [10] and ECSim 2.0 [11] codes, respectively.
57      The FANSY 1.0 code is a phenomenological hadronic interaction model implementing
58  quark-gluon string theoretical approaches and assuming various charm production cross sec-
59  tion parameters. In many features, it is close to QGSJET II model except for the $x_{Lab}$ spectra
60  of secondary particles including charmed ones, i.e., they appeared to be too soft as compared
61  to the LHC data).
62      Simulations of both experiments were performed by assuming that the darkening spots on
63  the X-ray films were created in interactions of cosmic ray hadrons, mainly by nucleons and
64  pions with energies $E_h \gtrsim 20$ TeV, produced by primary cosmic rays in the thick target of the
65  atmosphere above chamber (700 g/cm$^2$ and 600 g/cm$^2$, respectively, for Tien Shan and Pamirs
66  experiments). Consequently, it was assumed that the relative fractions of incident nucleons
67  and pions were 60% and 40% in the case of Tien Shan experiment, and 70% and 30% for the
68  Pamirs case while the indices of energy spectra for nucleons and pions were –3.10 and –3.22,
69  respectively, for both experiments. Angular distributions of incident hadrons were also taken
70  into account according to the experimental and simulation data. The Monte Carlo code ECSim
71  2.0 is based on GEANT 3.21 and allows to calculate the detector response for XREC of a given
72  design taking into account the exact experimental technique used in the "Pamir" experiment.

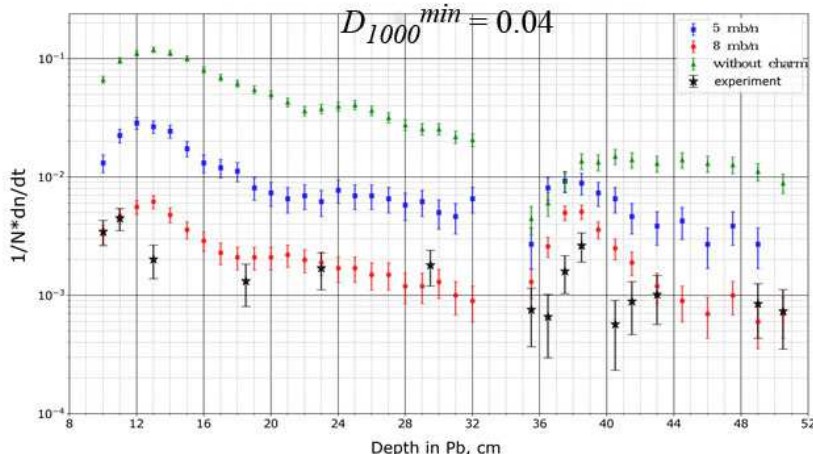

Figure 2: Comparison of experimental and simulated distributions of average number of darkening spots per a single X-ray film, over the depth $t$ (cm) of observation layers in the Tien Shan XREC (simulated distributions are calculated at three different $\sigma^{pr}_{pp \rightarrow c\bar{c}} \sim 0$, 5, and 8 mb/nucleon values and normalized to the total number of spots in XREC while the experimental distribution is normalized by the first three points of the corresponding simulated distributions).

## 3    New Experimental Measuring Technique

To analyze hadron absorption in lead calorimeters we used a new technique of XREC data processing. Conventional technique of the "Pamir" experiment includes reconstruction of hadronic cascades in the XREC applying the photometric procedures performed on microphotometers with diaphragms of radius $R = 84$ $\mu$m. The new technique is based on selecting and counting of only individual darkness spots on each film of a given observational layer by the naked eye enhanced only by a 2-fold magnifying glass. Advantages of the new technique proved by simulations:
• provides high sensitivity to determining of absorption curve parameters;
• increases the statistics of experimental data;
• does not implies conventional photometric procedures what is especially important given the uncertain sensitometric characteristics of new X-Ray films RT-6F used in this experiment;
• allows to avoid ambiguities related to reconstruction of hadronic cascades (more stable to systematic errors).

The performed analysis showed that the selection criteria for blackening spots in the experiment give the same result as for simulations if we select spots with optical density $D_{1000} \geq 0.04$ "measured" within a photometer diaphragm of radius $R = 1000$ $\mu m$. The estimated threshold energy of the recorded hadron-induced cascades is $E_{thr} \approx 6 - 8$ TeV.

## 4    Results and Discussion

To analyze experimental data we have plotted the darkness spot number distributions, normalized to the total number of spots per one X-ray film, over the depth $t$ of observation layers in both types of XREC, expressed in cm, for 3 optical density threshold values $D_{1000}^{min} = 0.01, 0.02$, and 0.04 and for three values of charm-production cross section, namely, $\sigma^{pr}_{pp \rightarrow c\bar{c}} \sim 8$, 5 and 0 mb/nucleon (0 means the absence of charm production). Although the selection criteria for experimental and simulated data were different (in the experiment, darkness spots were se-

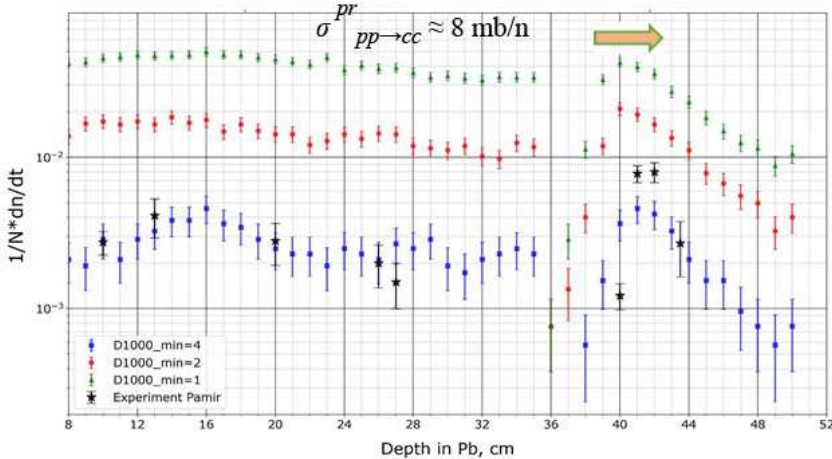

Figure 3: Darkness spot average number distributions, normalized to the total number of spots per a single X-ray film, over the depth $t$ (cm) of observation layers in the Pamirs XREC for three optical-density threshold values $D_{1000}^{min} = 0.01, 0.02, 0.04$ and at fixed value of $\sigma_{pp \to c\bar{c}}^{pr} \sim 8$ mb/nucleon.

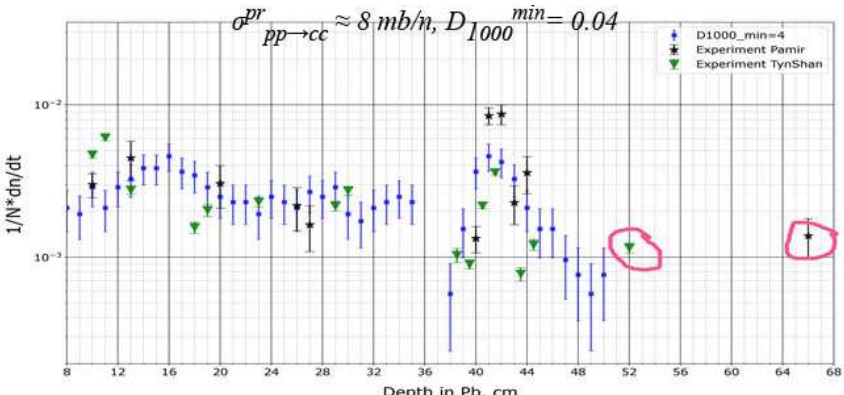

Figure 4: Normalized to the total number of spots, darkness spot average number distributions over the depth $t$ (cm) of observation layers, expressed in cm, in two-tier XRECs of both types as compared with simulation data at fixed values of $D_{1000}^{min} = 0.04$ and $\sigma_{pp \to c\bar{c}}^{pr} \sim 8$ mb/nucleon.

98  lected by naked eye enhanced by a 2-fold magnifying glass, while in simulation we applied the
99  standard for the "Pamir" experiment photomeasuring procedure with optical density threshold
100 $D_{1000}^{min} = 0.04$ simulations and our analysis showed that they are practically identical.
101     Fig. 2 demonstrates the sensitivity of two-tier XREC, namely, Tien Shan calorimeter to the
102 charm production cross section. This figure shows three sets of simulated data, calculated on
103 the assumption of different values of charm production cross section, and they are compared
104 with the experimental curve. In the case of high values of charm production we observe a peak
105 (bump) of electromagnetic origin at the hadron absorption curve which amplitude increases
106 with cross section. The best fit is in the case of $\sigma_{pp \to c\bar{c}}^{pr} \sim 8$ mb/nucleon. In the case of no
107 production of charmed particles, we do not see any bump at the absorption curve after the air
108 gap. More exactly, we see instead the deep dip but no bump at all that can be explained by
109 the hadron cascade decay in air gap and its further slow recovery in the low block of XREC.
110     Fig. 3 presents the comparison of three sets of simulation absorption curve data at different
111 thresholds of optical density $D_{1000}^{min} = 0.01, 0.02, 0.04$, provided $\sigma_{pp \to c\bar{c}}^{pr} \sim 8$ mb/nucleon, with

respect to experimental data measured with the naked eye from one-year exposure on at the Pamirs. You can see in this figure that the position of electromagnetic peak shifts to the right with the threshold of optical density $D_{1000}^{min}$ increasing so that maximum of simulated peak practically coincides with experimental one at $D_{1000}^{min} = 0.04$.

You can see in Fig. 4 the combined results of one-year exposure of two-tier XRECs at Tien Shan and Pamirs in comparison with model calculations assuming $D_{1000}^{min} = 0.04$ and $\sigma_{pp \to c\bar{c}}^{pr} \sim 8$ mb/nucleon. To compare distributions obtained using experimental XRECs of different design in the most adequate manner and, in particular, to compensate the difference in the depths of upper blocks, we have shifted the Tien Shan data for the lower block by 3 cm (the exact difference in thickness) that allows us to compare cascade developing in the lower blocks under equal conditions.

So, both distributions, as a whole, are in good agreement with each other, taking into account other smaller differences in the design of the two XRECs, different sensitivities of the films used, as well as different depths of the XREC location in the Earth's atmosphere (3,340 m and 4,370 m, respectively).

On the other hand, the obtained experimental distributions are also well reproduced by model calculations performed within the framework of phenomenological model of strong interactions FANSY 1.0 if we do not consider large depths.

In particular, taking into account the production of charmed hadrons, which effectively decay in the air gap between two lead blocks of the calorimeter through electromagnetic channels with the emission of electrons and gammas, allows us to qualitatively and quantitatively describe the experimentally observed peak on the absorption curves at a depth of $t_0 = 9$ r.l. in the lower lead blocks of the XREC. The amplitude of this peak, sensitive to the charm production cross section, makes it possible to conclude that it is as high as $\sigma_{pp \to c\bar{c}}^{pr} \sim 8$ mb at $\langle E_{Lab} \rangle \sim 75$ TeV and at $x_{Lab} \gtrsim 0.01$.

An unexpected result of both experiments is an excess of blackening spots at large depths of the lower lead blocks of the XRECs of both types (in particular, at $t_0 = 94$ r.l. and $t_0 = 119$ r.l., respectively). The corresponding points are marked in Fig.4 with red circles. In spite of low statistics and absence of data on neighboring layers, one can conclude that there exist a striking difference of experimental and simulated data on absorption of hadrons deep in the lead absorbers in both two-tier calorimeters (at the Tien Shan and Pamirs), namely at depths greater than 50 cm (we consider the total depth for both blocks). This discrepancy can not be explained by charm production.

To explain this significant excess of experimental spots we need to involve additional factors that may contribute to abnormally weak absorption of hadrons at great depths, for instance, the existence of strangelets emitted by strange quark stars, or beams of high-energy direct muons generated by charmed hadrons in the atmosphere. Another source of the excess of particle tracks at such great depths in calorimeters could be the production of dark photons in pPb interactions at TeV energies with their subsequent decay into $e^+ + e^-$ pairs. An appropriate experiment is currently being carried out at CERN by the NA62 Collaboration [12].

Surely, this result needs more careful study and analysis.

## 5   Conclusion

The nature and position of the features in the absorption curves, obtained in both experiments with a two-tier XRECs and as a result of model calculations, are in good agreement with each other, which indicates the correct interpretation of the experimental data as the observation of the birth and decay of charmed particles; Calorimetric experiments in cosmic rays with two-tier XRECs are rather sensitive to the cross section for charm production in the forward cone,

which is not accessible for observation in collider experiments.

Charmed-particle production cross section in the forward kinematic region $x_{Lab} \gtrsim 0.01$ is as high as $\sigma^{pr}_{pp \to c\bar{c}} \sim 8$ mb/nucleon at average hadron energy $\langle E_{Lab} \rangle \sim 75$ TeV (note that accounting for more realistic and hard $x_{Lab}$ spectra may decrease this value). The excess of hadron cascades in the depths of lead calorimeters can only be partially explained by the contribution of charmed particles. Additional sources of this excess may be strangelets emitted by strange quark stars, or dark photons, i.e., the carriers of a new fundamental force in the dark matter sector, or beams of high-energy direct muons generated by charmed hadrons in the atmosphere.

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
