# Peer review of "The Complex Nature of the Abnormally Weak Absorption of Cosmic Ray Hadrons in Lead Calorimeters at Super-High Energies"

_SciPost Physics Proceedings_

## Round 1 · Referee Report · Anonymous (Referee 1) · 2024-12-31

Strengths

1 interesting results about absorption of high energy showers in deep absorbers
2 data show sensitivity to charm production
3 the results are essentially clearly presented

Weaknesses

1 claim about dark photons not well supported
2 significance of the unexpected events not well explained

Report

The paper is well written and presents interesting results about charm production in CR induced showers at high altitude. The paper is considered acceptable for publication but requires some minor changes

Requested changes

line 51,52:
it is not understood the sequence of assembly affects the background. Please clarify the sentence

line 54:
find → found

line 60:
"they appeared..." : it is not clear whether this statement refers to FANSY and ECSim or to the other commonly used hadronic interaction models

Figures (quality):
They are compressed pixel files of very poor quality so that the text is hard to read. I strongly suggest to improve their quality to meet the standards of the Journal

Figure 2 and 3(contents):
It is not clear from the caption and text how to read the figures. Are the simulations with 0 and 5 mb scaled upwards for better readability? If not, it should be explained why a smaller charm cross section would lead to a higher number of spots.

line 79:
proved → have been proven

line 110...:
If I understand correclty, it is the shape of the curve in the second part of the calorimeter which matters, not the absolute values. If so, it is not clear to me why 8 mb is so much better than 4 mb. Can this be quantified or an uncertainty be assigned to the best cross section?

line 135-136:
There are data from e.g. the LHC about the charm production cross section.
For example: Eur.Phys.J.Plus 139 (2024) 7, 593 • e-Print: 2311.11426 also giving 8 mb at 5 TeV. A citation would be appropriate.

Figure 4 and discussion in line 137...:
The authors should state how many events are found in the last data points and it should also be stated over which range of depth these points are integrated. This is required before any statement about the significance can be made.

Recommendation

Ask for minor revision

---

## Editorial Decision

accepted_in_target_journal